# Wearables for Stress Management: A Scoping Review

**DOI:** 10.3390/healthcare11172369

**Published:** 2023-08-22

**Authors:** Maria Luisa González Ramírez, Juan Pablo García Vázquez, Marcela D. Rodríguez, Luis Alfredo Padilla-López, Gilberto Manuel Galindo-Aldana, Daniel Cuevas-González

**Affiliations:** 1Facultad de Ingeniería, Universidad Autónoma de Baja California, Mexicali 21280, BC, Mexico; maria.gonzalez@uabc.edu.mx; 2Laboratorio de Psicofisiología, Facultad de Ciencias Humanas, Universidad Autónoma de Baja California, Mexicali 21720, BC, Mexico; alfredopadilla@uabc.edu.mx; 3Laboratorio de Neurociencia y Cognición, Facultad de Ingeniería y Negocios, Universidad Autonónoma de Baja California, Mexicali 21725, BC, Mexico; gilberto.galindo.aldana@uabc.edu.mx; 4Instituto de Ingeniería, Universidad Autónoma de Baja California, Mexicali 21280, BC, Mexico; cuevas.daniel@uabc.edu.mx

**Keywords:** stress managing, interventions, wearables, smart garments

## Abstract

In recent years, wearable devices have been increasingly used to monitor people’s health. This has helped healthcare professionals provide timely interventions to support their patients. In this study, we investigated how wearables help people manage stress. We conducted a scoping review following the Preferred Reporting Items for Systematic Reviews and Meta-Analyses extension for Scoping Reviews (PRISMA-ScR) standard to address this question. We searched studies in Scopus, IEEE Explore, and Pubmed databases. We included studies reporting user evaluations of wearable-based strategies, reporting their impact on health or usability outcomes. A total of 6259 studies were identified, of which 40 met the inclusion criteria. Based on our findings, we identified that 21 studies report using commercial wearable devices; the most common are smartwatches and smart bands. Thirty-one studies report significant stress reduction using different interventions and interaction modalities. Finally, we identified that the interventions are designed with the following aims: (1) to self-regulate during stress episodes, (2) to support self-regulation therapies for long-term goals, and (3) to provide stress awareness for prevention, consisting of people’s ability to recall, recognize and understand their stress.

## 1. Introduction

Stress occurs when a person perceives a stimulus as a threat, activating their autonomic nervous system and releasing hormones like adrenocorticoids, glucocorticoids, catecholamines, and growth hormone, to mention a few [1]. These hormones have diverse effects on the body, including increased heart rate, muscle tension, blood pressure, and breathing frequency [2]. Due to the above, stress may disrupt homeostasis (the balance required for the human body to function properly) [3], contributing to several health problems such as arterial hypertension, heart disease, abnormal sleeping patterns, depression, and anxiety [4,5,6]. Furthermore, it alters and distorts social relationships, sometimes leading to work absenteeism, drug addiction, personality disorders, and even suicide [7,8].

To diagnose stress, mental healthcare specialists have explored individual and combined methods, including measurement of the cortisol hormone from blood or saliva samples, monitoring of physiological signals (e.g., Heart Rate Variability and galvanic skin response) [9], and the application of validated questionnaires (e.g., Perceived Stress Scale (PSS) [10]). Once stress is detected, relaxation techniques are suggested, such as taking deep breaths, listening to music, meditating, exercising, and eating well [11,12,13]. However, they are applied when people are overly stressed and have some overt health problems [14].

Thus, the main challenge is detecting stress in time to be treated during the early stages without requiring people to attend a laboratory to use specialized clinical equipment or to answer validated questionnaires when the stressful events have passed [15]. The research community has studied how mobile technology can be a suitable alternative to monitoring early stress manifestations and provide interventions anywhere and anytime [16,17]. According to [18], “an intervention is the manipulation of the subject or subject’s environment to modify health-related biomedical or behavioral processes”. Examples of interventions are drugs, devices, and strategies to change health-related behaviors or prevent health conditions. According to a report by the World Health Organization (WHO), during the COVID-19 pandemic in 2020, telemedicine and teletherapy, including mobile health technologies, played a positive role in 80 percent of developing countries that used them to bridge gaps in mental health [19,20].

Wearable devices (or wearables) are an example of a mobile technology that people adopt to monitor their health and well-being [21]. These devices are worn on specific body parts (e.g., wrist, hand, neck), as they have sensors that continuously measure physiological signals, like heart rate, temperature, and galvanic skin response, to mention a few [22]. The main features of wearables are that they can be connected to the internet to transmit, log, or analyze data. Also, they can be linked to other electronic devices to extend their functionalities; for instance, smartwatches have traditionally been designed to monitor users’ performance during sports activities, which can be viewed from purpose-specific smartphone applications. Nowadays, wearables are also used to manage health since they incorporate diverse, smart sensing and communication capabilities, which attract consumers and pave the way for market growth [23]. Statistics show that in 2021, eyewear or headwear devices occupied about 31.5 percent of the wearables market, while watches held 30.5 percent [23,24]. However, by 2030, is expected that wristwear devices will dominate the market, followed by eyewear and headwear, footwear, neckwear, and others [23,24].

Furthermore, research has focused on enhancing wearables’ sensing capabilities through machine learning algorithms to detect stress [25,26] and design interventions, such as guiding people to take deep breaths [27]. In the state of the art, some reviews aimed to collect research to analyze how novel wearable devices have been used to detect stress. For instance, Hickey et al. [17] identified that smart bands, smart watches, and headbands are the most used to estimate stress by analyzing physiological data such as Heart Rate Variability (HRV) and Heart Rate (HR). The authors also found that the average HR used by many commercially available devices is less accurate in detecting stress than HRV, electrodermal activity, and respiratory rate. Similarly, another review identified that HR is the most precise biosignal to detect stress, in addition to galvanic skin response, and that the most preferred sensing platforms for data collection are Empatica (wristwear), Emotiv (headwear), and Shimmer (bodywear) [16]. It also reports that the most explored machine learning algorithms for mental stress detection are Fuzzy Logic and K Nearest Neighbors (KNN). Fuzzy Logic algorithms achieved the highest classification accuracy (96.16%) with decision trees, followed by Logistic Regression, Linear Discriminant Analysis (LDA), and multilayer perceptron [16]. Finally, a review of smartphones and wearable devices reports the combinations of stress signals and machine learning models explored for predicting stress [28]. They found that using Electro-Dermal Activity (EDA) and HR combination yields the best results with an accuracy of around 95% by using either LDA, Support Vector Machine (SVM), kNN, or Fuzzy Logic. Notably, this is the only review that presents an overview of smartphone apps designed to relieve stress; however, it does not analyze their efficacy.

We conclude that the effect of wearable-based approaches on alleviating or reducing stress has not been analyzed. Previous reviews [16,28] have focused on presenting overviews of wearable devices, including those based on commercial platforms, machine learning algorithms, and physiological data used to detect stress levels. Therefore, the limitations and open research opportunities for wearable-based interventions have yet to be discussed. Further investigations are needed to understand the current research on using wearables to deal with stress.

A scoping review is a study conducted to examine emerging evidence from a body of literature on a given topic [29]. It helped us answer the following research question: how do wearables help people manage stress? To address it, we (i) identified the technological characteristics used for deploying interventions to manage stress, (ii) extracted data related to the assessments of the proposed interventions, and (iii) classified the interventions based on their aim to manage stress. We obtained three types of intervention aim: (1) self-regulation during a stress episode, (2) self-regulation therapies, and (3) awareness for prevention. Our work aims to present an overview of studies presenting designs of wearable-based interventions and evidence of their benefits in managing stress. To this end, we carried out a scoping review of studies reporting user evaluations of wearable-based strategies to manage stress. The type of studies included in the review are those presenting evaluations associated with the development life cycle of interactive health systems, as explained by Yen and Bakken [30].

## 2. Materials and Methods

This scoping review was conducted in accordance with the Preferred Reporting Items for Systematic Reviews and Meta-Analyses extension for Scoping Reviews (PRISMA-ScR) [31].

### 2.1. Search Keywords and Databases

We identified relevant studies in the IEEE Xplore, PubMed, and Scopus databases. We performed the search on 8 March 2022, and updated on 26 May 2023, using terms related to (1) *wearables*: body-worn garments, smart textiles, wearable sensors, wearable systems, wearable, and garment; and (2) *the aim of using wearables:* stress, burnout, distress; stress management and stress monitoring. These terms were used to create generic search strings using the Boolean AND and OR operators, as the following: (“body-worn garments”) AND (stress OR “stress management” OR “stress monitoring” OR burnout OR distress). Seven generic search strings were generated and adapted to the databases following their guidelines. As explained in the next subsection, the adaptations included setting filters for retrieving documents that met the inclusion criteria related to language, publication dates, and document type.

### 2.2. Inclusion and Exclusion Criteria

The inclusion criteria regarding the publication characteristics were studies written in English and published in journals or conference proceedings between 1 January 2009 and 31 December 2022.

To define the type of studies to include in this review, we used the usability specification and evaluation framework for health information technology reported by Yen and Bakken [30], which specifies that an interactive health system can be incrementally evaluated through five types of studies: (1) analyses to identify users’ needs and propose initial system’s requirements; (2) lab sessions to assess system performance; (3) lab sessions to evaluate user–system interaction performance; (4) user’s assessment of system’s usability quality aspects, such as learnability and satisfaction; and (5) user’s evaluation of the system’s impact on health related-outcomes.

Based on the above, we included studies of types 3, 4, and 5 in this review, requiring users to interact with the proposed wearable technology. These studies report the impact of technology on health-related outcomes or usability-related outcomes such as users’ engagement and awarenesss. Therefore, we excluded studies that only focused on: intervention designs, performance evaluations of a method for detecting stress, such as evaluations of machine-learning-based methods, and literature reviews.

### 2.3. Study Selection

The study selection consisted of three stages: identification, screening, and inclusion (see Figure 1). The identification stage consisted of searching for relevant studies and retrieving their metadata in RIS format to be uploaded to the Rayyan software (https://rayyan.qcri.org, accessed on 24 June 2023), a collaborative tool to facilitate systematic reviews. We used it to eliminate duplicates, review titles and abstracts during the screening phase, tag the studies to differentiate between included and excluded, and describe our reasons for excluding studies [32]. After the elimination of duplicates, the screening stage was performed. This consisted of checking that the title and abstract of the selected studies answered the following questions about the inclusion criteria addressed: (1) Is the study related to stress? (2) Does the study use a wearable device? If both answers were affirmative, the study was assessed for eligibility, which consisted of reading the full text to determine if it met the inclusion criteria. Finally, data from the included studies were extracted for further analysis.

Two co-authors (M.L.G.R. and J.P.G.V.) performed the identification and selection stages. The inclusion stage was performed by three co-authors (M.L.G.R., M.D.R., and J.P.G.V.). Any disagreements were resolved through discussion among the co-authors. The results of each stage were presented to the rest of the co-authors for their validation.

### 2.4. Data Extraction and Analysis

We followed deductive and inductive approaches to identify the data about the studies’ characteristics [33,34]. To this end, we predefined a set of data categories deductively, i.e., based on existing concepts and knowledge obtained from the literature on the subject of this review [16,28,29]. Using this approach, we obtained information about the intervention, such as physiological signals measured to detect stress (e.g., ECG, EEG, HR). The form factor of wearable devices refers to the physical characteristics of the device or object, such as its dimensions and shape [35], (e.g., smart watch, smart band, smart glasses), body parts where devices were worn (e.g., wrist-wear, torso-wear), hardware and software trademark for commercial devices or if they were custom-made, and the interaction modality supported to present information to users for managing stress (e.g., visual, auditory, or tactile). We also obtained data about the context of the study related to where the study took place (e.g., school, hospital, building); who participated in it (e.g., the type of participants, such as students, veterans, and older adults); what assessments techniques were used, such as if experiments were conducted under controlled conditions; stressors used to induce the stress; and validated instruments to measure participants’ stress levels. Furthermore, we extracted text reporting the most significant results and conclusions the articles’ authors reported about stress.

On the other hand, some data types emerged during the extraction while reading the articles, i.e., they were identified inductively. One of them was the aim of the intervention. We analyzed which interventions had the same purpose based on the definition of stress management interventions, which refers to activities the affected person performs, commonly accompanied by a health care specialist, to improve their well-being and reduce stress, address the causes of stress, or reduce the impact of stress [36]. As a result, we identified three main types of intervention supported through wearables. We discussed them with the co-authors of this review, experts in Psychology, resulting in two categories related to self-regulation and one to prevention. Self-regulation refers to a person’s ability to control their emotional and behavioral responses to stressful situations [37]. Related to this, we identified studies that aim to help a person self-regulate during a stress episode. These study types provide interventions when a stress episode is detected. The second category was self-regulation therapies, which aim at a person’s ability to regulate emotional, cognitive, and behavioral responses based on long-term goals [38]. The third category was stress awareness for prevention, consisting of people’s ability to recall, recognize and understand their stress [14]. Similarly, we analyzed the results and conclusions texts extracted from each article to identify if the authors found a positive effect of the intervention on primary health outcomes and which secondary outcomes related to the wearables’ usability were assessed, such as satisfaction and user experience.

Finally, we extracted publications’ characteristics of the studies, such as the year of publication and type of document, i.e., an article published in a journal or conference proceedings.

Two co-authors, M.L.G.R. and J.P.G.V., participated in the extraction stage. We generated an online spreadsheet using Google Sheets [39] to make it easier to independently extract information from the studies and collaborate to resolve disagreements. The Google spreadsheet was extended when data types were identified inductively. This required an iterative review of the set of studies. The results were discussed with M.D.R. for their validation and to generate the final discussion.

## 3. Results

### 3.1. Search Result

As illustrated in Figure 1, the search yielded 6259 studies, of which 1613 were duplicated. After screening the titles and abstracts, 103 studies were selected for full-text reading. A total of 40 articles met the inclusion criteria from which we extracted data.

### 3.2. Publication Characteristics

Figure 2 illustrates a stacked bar chart that depicts the number of conference and journal articles identified per year. The graph reveals an upward trend in the number of studies over the last four years of the analyzed period (2019–2022), during which more than half of the studies (N = 25, 62.5%) were published. Furthermore, there were more studies published in journals (N = 22, 55%) than in conference proceedings (N = 18, 45%). Notably, journal publications dominated in the last two years compared to previous years (N = 11).

### 3.3. Characteristics of the Intervention

#### 3.3.1. Wearable Devices

Smart wearables are electronic devices equipped with wireless sensors that are integrated into clothing or accessories [40]. Figure 3 displays different types of wearables that have been identified in the studies selected for this scoping review. These wearables are depicted using icons aligned with specific parts of the human body (i.e., the smartband on the wrist). Each icon is accompanied by a number to the right, within parentheses, representing the frequency with which the device has been identified in the studies. In Figure 3, it can be seen that wrist-worn devices (N = 35, 87.50%) are the most commonly used among users [26,41,42,43,44,45,46,47,48,49,50,51,52,53,54,55,56,57,58,59,60], followed by devices worn on the torso (N = 19, 47%) [25,42,59,60,61,62,63,64,65,66,67,68], which include nine smart chest bands, five smart pins, and five patches; the head (N = 3, 7.5%) [62,69,70]; and the eyes (N = 2, 5%) [46,47], i.e., smart glasses; the neck (N = 2, 5%) [71,72], including a scarf and necklace, and the arm (N = 2, 5%) [73,74]; and the fingers (N = 3, 7.5%) [49,68,75]. We found nine studies (22%) that used a data acquisition system with more than one sensor worn on several body parts; these systems do not comply with the definition of wearable [41,45,46,47,48,49,59,70,76].

We identified 21 studies (52.5%) that report the use of commercial devices, such as Empatica E4, Spire Stone, Stila and Apple Watch, and Doppel [25,26,42,46,47,50,51,53,54,55,56,58,59,60,61,63,64,65,69,70,77]. Only 11 of them used custom-made software [14,46,47,48,50,56,58,63,64,70,78], 4 used commercial software (e.g., Breathe, Lief app, Stila, Wahoo) [25,51,61,77], and 5 do not report whether using commercial or custom-made software [65,69,72,74,76]. Moreover, 11 studies (37.9%) reported using a custom-made wearable device [41,43,44,45,52,62,67,71,72,73,76]. For instance, Wu et al. [76] used ECG and respiration sensors to build an HRV biofeedback system for stress reduction and autonomic nervous system modulation through resonance frequency respiration training.

#### 3.3.2. Stress Monitoring

The studies used opportunistic and participatory sensing paradigms for collecting data from users [79]. In opportunistic sensing, data are collected automatically from sources associated with specific stress symptoms, such as difficulty breathing detected through physiological signals or user behaviors inferred from inertial data. The physiological signals have been the most used, depicted in Figure 4. It shows that 24 studies (60%) reported using Heart Rate (HR) and Heart Rate Variability (HRV) measured from ECG or PPG sensors [26,41,42,43,45,46,48,51,52,55,56,59,60,61,62,63,64,67,68,70,72,76,78]. Other physiological signals reported are respiration rate (N = 5, 12.5%) [25,63,74,76,77], brain activity monitored with EEG sensors (N = 4, 10%) [62,65,69,70], galvanic skin response (N = 13, 32.5%) [14,26,42,44,48,49,56,57,58,59,68,73,75], temperature (N = 6, 15%) [43,49,53,59,63,73], oxygen saturation (SpO2) (N = 2, 6.8%) [43,44], and stress (N = 1, 2%). Furthermore, some studies propose other types of information such as inertial data (N = 8, 20%), for instance, monitoring users’ walking behaviors [45,66], arousal [53], outburst patterns from their movement [64], sleep [52] and posture [63].

In the participatory sensing paradigm, the interventions include apps that ask users to provide their stress self-perception by presenting questionnaires. In this sense, four studies (10%) report apps presenting questions to assess users’ stress [41,45,52]. For example, Edirisooriya et al. [45] present a system that combines opportunistic and participatory paradigms. It recognizes a person’s stress level by monitoring heart rate, counting steps taken, and using a simple questionnaire. Based on these data, it provides the user with a virtual environment suitable for mindfulness-based activities [45]. Zhang et al. [41] present a stress detection algorithm to obtain data from a PPG sensor and the subject’s physical activity to detect users’ stress levels. Afterward, the system launches a questionnaire that users should complete to determine their stress self-perception. This information is used to provide the intervention. Finally, Ponzo et al. [52] present a system that estimates stress levels by monitoring physical activity, sleep quality, heart rate, and self-perception via an ecological momentary assessment, EMA. The system provides exercises (e.g., deep breathing and relaxation techniques) according to their self-perception.

#### 3.3.3. Interaction Modality

Four modalities were used to present information to the users that would make it easier for them to follow the stress management technique supported by the intervention: V (Visual), A (Auditory), T (Tactile), and O (Olfactory). Table 1 presents four columns. The first column contains the study references. The second column specifies the interaction modality used to inform individuals about their stress levels. The third column displays the interaction modality employed to provide stress management interventions. Finally, the fourth column describes the intervention supported by wearable devices. In Table 1, we can identify the following findings: visual modality is the most used to provide stress feedback level (N = 26, 65%) [25,26,41,44,45,47,52,54,55,56,57,58,62,63,64,66,69,70,74,76,77,78], followed by the auditory modality (N = 11, 27.7%) [43,44,45,46,47,52,54,69,70,71], tactile (N = 10, 27.58%) [42,44,54,55,61,65,71,72,74] and olfactory (N = 3, 3.44%) [52,67,78]. For instance, BreatheWell Wear is an app running on mobile phones of users wearing smartwatches [54]. The app guides users to pace breathing by displaying (V) their heart rate in real-time. Thus, users should pace their breathing by matching it to the movement of the blue bar. The wearable vibrates (T) at the end of each inhalation and exhalation when the visual pace bar reverses the direction of movement for the current breathing cycle segment. The app also offers several auditory (A) guidance options, listening to calming sounds and music.

Only two studies do not report an intervention encouraging users to follow relaxation activities like the ones above. Instead, they describe interventions designed to give feedback about users’ stress levels [51,73]. In this sense, 20 (50%) studies, including the two mentioned previously, describe interventions designed to provide stress level feedback [25,26,41,51,54,56,57,58,61,62,63,64,70,72,73,74,75,76,77]. As shown in Table 1, they all used the visual (V) modality. For instance, Chen et al. [63] represent the stress level and how it changes over time through a 2D color visualization graph, and 2 of the 16 studies combined visual feedback with auditory (A) feedback [72] or tactile (T) feedback [74].

#### 3.3.4. Intervention Time Length

In the studies, it was identified that the interventions are evaluated with different time periods. It was identified that most of the studies report interventions performed daily during weekly (N = 13, 32.5%) [45,46,47,51,52,54,61,62,66,68,70,74,76].

### 3.4. Study Method

#### 3.4.1. Study Context

We identified the studies reporting experiments carried out under controlled conditions. These experiments are characterized by being conducted within a laboratory or through field observations, in which participants are requested to perform specific evaluation tasks to interact with an intervention [80]. In some of these studies, the participants’ stress was generated through external stimuli known as stressors tests. Experiments carried out under controlled conditions were the most reported evaluation techniques of wearables (N = 22, 55%) [42,43,44,45,46,47,48,50,54,56,58,59,63,64,65,67,68,69,70,72,75,76].

From the 40 studies, 18 (45%) were conducted in a school environment, e.g., a classroom or office [14,44,45,46,47,50,52,55,56,58,64,67,70,74,75,76,77], and the participants were students from different educational levels, such as elementary [70], university [14,44,45,46,47,50,52,55,56,58,64,67,74,75,76,77], and postgraduate [26,67]; three (N = 5, 12.5%) were conducted in a work environment [25,41,60,73,78]: one (N = 1, 2.5%) with veterans [54], one inside a clinic (N = 1, 2.5%) [68], one in a shelter (N = 1, 2.5%) [57], one in the forest (N = 1, 2.5% ) [49], one (N = 1, 2.5%) with unemployed participants [76], and one (N = 1, 3.44%) with elderly participants [43]. The rest of the studies do not define where they took place (N = 9, 22.5%) [48,59,66]. On the other hand, most of the studies recruited adult participants in the age range of 18 to 64 years old (N = 34, 85%), followed by children aged 1–12 years old (N = 2, 6.89%) and other age groups (elderly and teenagers).

Participants were subjected to stressor tests only in experiments with controlled conditions, which were carried out in a closed area, such as a school or office. The most used stressors tests were academic-related tasks [42,45,56,58], Stroop Test [47,67], Sing a Song Stress Test [72] and video games [48,65].

Table 2 exhibits two columns. The first column comprises the names of the instruments or scales utilized to measure the level of stress or anxiety in the study participants. The second column presents the studies in which each instrument has been applied. Based on the data in Table 2, we have identified the following findings: out of the 25 studies, validated instruments were employed in assessing the subjects’ self-perceived stress. The State-Trait Anxiety Inventory (STAI) (N = 8, 20%) emerged as the most frequently used instrument, followed by the Perceived Stress Scale (PSS) (N = 6, 15%).

#### 3.4.2. Study Outcomes

Most studies (N = 31, 77.5%) present positive outcomes on how their interventions help people manage stress [25,26,41,42,44,45,47,49,50,51,52,53,54,56,60,61,62,63,64,65,69,70,71,72,75,76,77], for which they provide evidence regarding changes in physiological signals [26,61], stress self-perception [67], awareness of stress level [53,65,71], and metrics associated with usefulness [54], user engagement [51] and task efficiency [51]. However, two studies do not report stress reduction [55,74]. The authors conclude that some limitations of their methodology could cause this result.

### 3.5. Wearable-Based Interventions to Manage Stress

The following shows how the studies were classified into three categories according to the purpose of the intervention supported through wearables. Table 3, Table 4 and Table 5 present the studies we have identified within the established categories. Each table provides relevant information about the wearable devices used, the participants involved, the study context, the implemented intervention, and the results obtained from said intervention.

#### 3.5.1. Self-Regulation during a Stress Episode

To self-regulate during a stress episode, individuals must be aware of at least one response of their physiological signals associated with stress (e.g., HR, Temperature, Sp02, GSR) [81]. We found 24 studies supporting auto-regulation (see Table 3); all of them describe the use of wearable devices to monitor and compare stress-related physiological signals to determine whether they meet normality thresholds [25,26,41,42,43,44,45,46,47,48,49,50,60,61,62,63,64,65,69,71,72,73,74,75,76] To this end, mathematical functions [41,42,43,44,45,46,47,48,49,50,60,61,62,63,64,65,69,71,72,73,74,75,76] or artificial intelligence algorithms [25,26] were used. An example of this type of management intervention is presented by Yamane et al. (2021 [55], in which two wearables are used—a patch-type ECG sensor to measure heart rate to detect stress episodes, which launches the intervention in the Apple smartwatch. It consists of the Breath app showing an animation to guide people to take a deep breath.

Several studies highlight the importance of personalizing interventions. One of the studies argues that auto-regulation can be tailored to the context subject [26]. It proposes a wearable system in that the people’s location is analyzed to tailor the intervention to their needs, i.e., the system identifies whether the people are physically active and in a free context, such as a weekend or holiday, to suggest traditional relaxation methods, such as yoga. This work argues that technological-based relaxation methods may be appropriate when people are physically inactive and in a restricted environment, such as at work or in an office.

On the other hand, three studies argue that the intervention can be tailored to the subject’s characteristics, such as health condition [42,54,64]. For example, Torrado et al. (2017) describe the design of interventions to help subjects with autism spectrum disorders control their moods and behaviors [64]. Likewise, Morris and Wallace (2018) present the design of an application for Android Wear smartwatches to assist military service members with post-traumatic stress disorder and traumatic brain injury to use deep, slow diaphragmatic breathing to manage stress [54]. Furthermore, in [42], the intervention consists of heartbeat-like vibration on the wrist to the rhythm of the subject’s heart rate. Finally, one study reports involving a care network member to personalize the intervention considering the preferences of the patient using the technology [64].

#### 3.5.2. Self-Regulation Therapies

In this category, we identified seven studies [54,55,56,57,67,68,70]. They are characterized by supporting behavioral therapies such as Cognitive Behavioral Therapy (CBT) [82]. For example, Skulimowski and Badurowicz (2017) use horticulture therapy, which consists of taking care of a bonsai in a virtual reality game environment [56]. Users must water, dust the leaves, and prune the virtual bonsai daily. At the same time, some physiological signals are measured to detect if users are stressed, which impacts the health of their bonsai since it starts to grow slowly and wither. Another example is presented in [70]. This work presents several video games that attempt to replace negatively conditioned stimuli with positive ones to help change negative thought patterns [82]. The video games use metaphors to help users learn to control the body’s physiological responses to achieve relaxation. For example, exhaling slowly and calmly and blowing out slowly to ignite a flame. Likewise, Breathewell is an app that runs on a smartwatch to allow people to set reminders to perform breathing exercises guided by music and visualizations [54]. Its purpose is for the subject to perform regular breathing exercises that allow self control when facing stressful episodes [54].

#### 3.5.3. Awareness for Prevention

The objective of the intervention is to provide persons with awareness of the daily activities that trigger their stress. To this end, daily or historical information regarding stress is presented to help them to make decisions to change their lifestyle. In this category of interventions, nine studies were identified [25,51,52,53,58,59,66,77,78]. All studies use dashboards, i.e., a kind of “summary” that collects data from different sources and presents it in a way that is easy to understand at a glance. The dashboard can be updated daily or weekly [25,51,52,53]. For instance, Wang et al. [51] present a Wear OS smartwatch equipped with the Stila smartwatch application as a pulse rate provider. The compound Stila Computed Stress graph and activity list were designed to encourage users to compare their computed and perceived stress levels and relate these to their daily activities, thus fostering their stress self-regulation. Another example is presented in [53], where galvanic skin response is collected using a DTI-2 wristband and stored in digital calendars, like LifelogExplorer, to provide comprehensible interactive visualizations of users’ arousal information in the context of their weekly life events. This enables users to learn about their stress behavior patterns and to decide which are relevant and can be changed. Finally, two studies used numerical data or text to represent the information associated with stress levels and the relationship with the activities performed [51,77]. For example, Van et al. [77] present the use of the Spire stone sensor to monitor subjects’ breathing, which is classified into patterns, such as calm, concentrated, tense, neutral, or active. Then, they are associated with their cognitive or emotional states during their regularly scheduled activities on school days. Through a dashboard on the Calm application, subjects can visualize the patterns they have presented and for how long throughout the day.

## 4. Discussion

### 4.1. How Do Wearables Help People Manage Stress?

The use of wearable-based interventions helps to reduce stress since most studies report positive outcomes [25,26,41,42,44,45,47,49,50,51,52,53,54,56,58,59,60,61,62,63,64,65,66,68,69,70,71,72,75,76,77]. One of the main benefits of using these technologies is that people can receive assistance anywhere and anytime through natural interactions, such as haptic interaction [44,61]. Three strategies for stress management predominate, where the self-regulation during stress episodes strategy is the most explored [26,41,42,43,44,45,46,47,62,63,65,69,71,72,73,74,76], followed by stress prevention and self-regulation therapies [54,55,56,57,67,68,70].

The studies provide promising results regarding stress management through wearable devices. However, their results cannot be generalized to the rest of the population because most were conducted with students under controlled conditions in academic settings [44,50,55,56,58,64,67,75]. Furthermore, there is a lack of evidence on adopting wearable devices for stress management. Therefore, more studies are needed to understand the barriers to adopting this technological approach to cope with stress, such as privacy and intrusion issues. The advances in research using body-worn garments and wearables from recent years have permitted essential findings in different fields. For example, for managing recovery in sports [83,84,85]. However, emerging studies share a focus on healthcare, particularly, regarding the relationship between Heart Rate (HR) and stress, which is the main interest of our study. Wearables offer a great advantage in large periods of time-monitoring of HR in natural environments, such as work, and signal processing, which leads to the possibility of understanding the effects of chronic stress on HR during circadian periods [86]. Dealing with stress is an everyday challenge for most people, and people face particular conditions and situations which lead to the need for specific features for different types of support.

According to this scoping review, the research has gained strong consistency between the needs for the wearable’s design and the requirements of the users; previous research addresses how technological skills may be developed to create partnerships that take into account the person, the situation, and the right kind of support delivered by smart wearables [87]. In general, health disciplines, particularly mental health, require feedback for assessing the continuous effects of treatments on the mental health of individuals. The findings from studies suggest that when monitoring therapies with wearable devices, participants show 15.8% fewer negative episodes of stress, 13.0% fewer distressing symptoms, and 28.2% fewer days feeling anxious or stressed after the 4-week intervention period [25].

On the other hand, this scoping review leads to identifying that wearables, when used with common objectives with mental health disciplines, are capable of providing relevant insights about the quality of life; it is demonstrated that they strongly help to monitor daily life activity. According to a systematic review, the upcoming efforts on improving the efficiency of wearable outcomes for concerning health should focus on elongating the 24 h physical behavior construct, as well as looking for standard protocols that are integrated into a validation framework [88]. Artificial intelligence embedded in smart wearables still requires developing reliability for decision making on physiological-stress-related information classification. However, studies at the edge of this systematic review demonstrate good parameters, such as cross-validation accuracy of 99.7%, sensitivity of 100%, precision of 97% [89], and a scientific background on heart illness prevention validation protocols [90].

### 4.2. Opportunities for Future Research

Since we have identified few studies that explored prevention and self-regulation strategies, there is an opportunity to investigate and develop comprehensive solutions that support the three strategies reported in this review. In this way, technologies would accompany people to teach them to manage and prevent stress and help them when facing a stress episode. To reach this end, designers may consider using the ”technology as a partner” framework to develop wearable-based interventions [87]. This framework proposes designing wearable devices that act as partners, either as (1) a therapist helping people manage their stress, (2) as a human interpersonal association that would be part of a care social network of the affected person, and (3) as a partnership with a pet, where pets provide companionship, care, and comfort.

The evaluations of the interventions have been carried out in laboratory settings under controlled conditions, so more evidence gathered in a natural context is needed to conclude about the benefits of these approaches in the long term and their adoption. Also, many studies were excluded due to not reporting intervention evaluations, which indicate that the research interest is growing. Among these studies, several were recent publications presenting designs of wearable devices embedded in clothing, i.e., wearable garments [91,92,93,94,95]. Similarly, we identified some preliminary research works proposing novel technologies to detect other stress information sources, such as cortisol and repetitive body movements in legs and fingers [96]. Therefore, further reviews may be needed in the near future to map the research on novel developments in this topic.

Finally, from the psychological perspective, three elements must be addressed to manage stress: (1) monitoring of physiological signals, (2) self-perception, and (3) assessment by an expert. However, few studies address all three elements.

### 4.3. Limitations

This paper presents a literature mapping to understand how wearable devices may help people manage their stress. Therefore, the results are presented descriptively, and no statistical analysis or critical evaluation of the findings was conducted.

Our scoping review did not comprise an assessment of the methodological quality of the studies. Therefore, studies of different quality levels were included.

Finally, although an exhaustive literature search is attempted, ensuring that some relevant studies have not been overlooked can be difficult. This may be due to time constraints, limitations in the databases used, language barriers, or difficulties in accessing certain types of literature, such as unpublished reports or ongoing studies.

## 5. Conclusions

Wearable devices have been recognized as vital tools for detecting stress episodes and offering interventions for its management. These interventions based on wearables have shown promising results in effectively managing stress. Our approach to organizing the studies has shed light on the fact that these interventions were primarily designed for self-regulation during stress episodes, self-regulation therapies, and raising awareness for stress prevention. However, it is essential to acknowledge that the generalizability of the results might be limited, as the evaluation of wearables was conducted in a specific context, particularly within an academic environment, and under controlled conditions. Consequently, it is imperative to conduct more extensive evaluations in real-life and daily settings to assess the broader applicability and effectiveness of these wearable-based interventions in diverse populations and various stress-inducing situations.

Further, while the majority of wearable studies have focused on smartwatches and activity bands, it is worth noting that ongoing research is exploring innovative avenues, such as smart garments and sensing technologies, to detect stress through cortisol level analysis. This evolving trend indicates that the future of wearables will likely involve seamless integration into people’s everyday routines, making stress management more effortless and user-friendly.

The growing interest in smart garments and cortisol-based stress detection unveils exciting possibilities for the next generation of wearables. These advancements hold the potential to provide even more accurate and personalized stress management solutions, catering to individual needs and preferences. By seamlessly integrating wearable technology into daily activities, individuals can monitor and respond to stress in real-time, fostering a proactive approach to maintaining mental well-being.

Moreover, the development of wearables with advanced stress detection capabilities could extend beyond individual benefits. Researchers and healthcare professionals might harness the data collected from these devices to gain deeper insights into stress patterns at a broader societal level. This information could facilitate the implementation of targeted stress management programs, fostering healthier communities and workplaces.

In essence, the ongoing shift towards exploring smart garments and cortisol-based stress detection signifies a promising future for wearables, where they become indispensable companions for managing stress in our fast-paced lives. As technology continues to evolve, these wearables are poised to play an increasingly significant role in supporting mental health and overall well-being, empowering individuals to lead healthier, more balanced lives.

## Figures and Tables

**Figure 1 healthcare-11-02369-f001:**
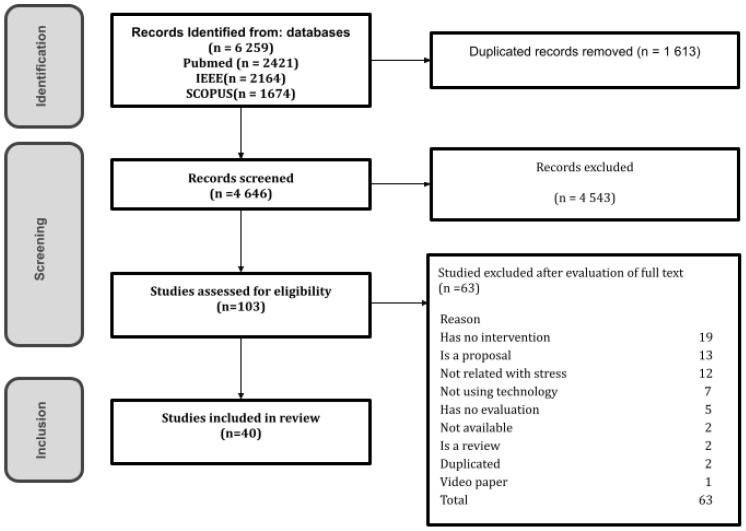
Study selection process.

**Figure 2 healthcare-11-02369-f002:**
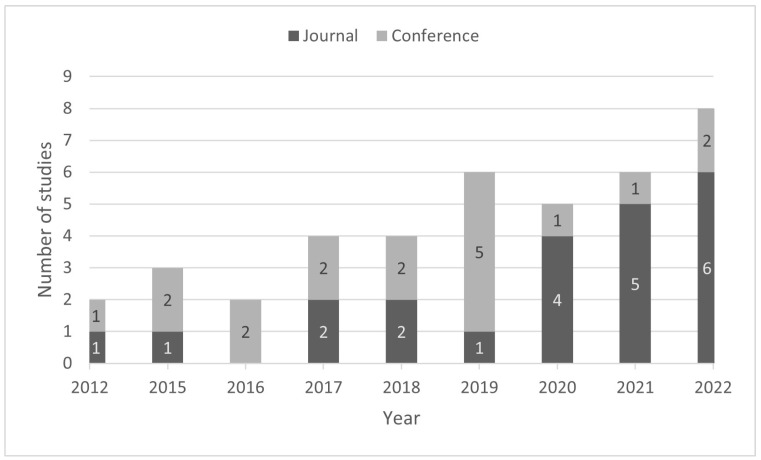
Type of documents published between 2012 and 2022.

**Figure 3 healthcare-11-02369-f003:**
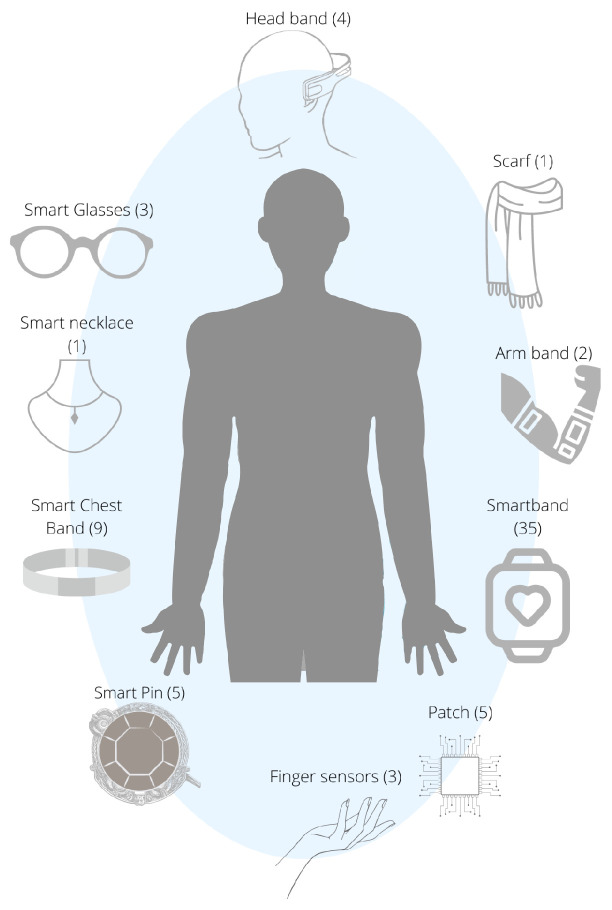
Form-factor of wearables devices used to measure physiological signals.

**Figure 4 healthcare-11-02369-f004:**
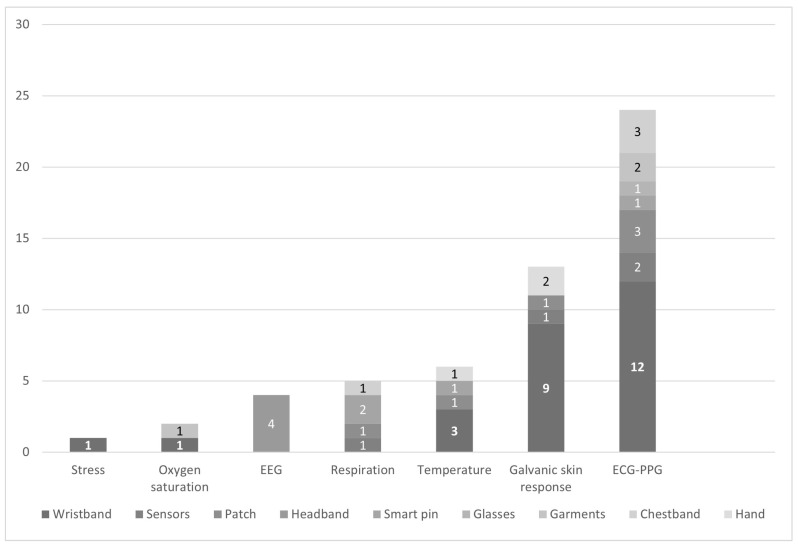
Wearable devices used to gather physiological signals.

**Table 1 healthcare-11-02369-t001:** Interaction modalities used in the intervention.

Paper Id	Stress Level Feedback	Interaction Modality	Intervention Description
[44]	VA	V A T	Applies stimuli based on heat, cold, vibration, ambient light, and sound
[63]	V	V	YOGA breathing exercise: inhale, hold, exhale
[61]	V T	T	Breathing exercise, vibration and visual cues
[73]	V	NA	Meditation
[74]	A T	V	Deep breathing
[26]	V	V	Mindfulness and Yoga
[41]	V	V	Guided breaths
[76]	V	V	Guided breaths
[54]	V	V A T	Breathing
[55]	NA	V T	Breathing
[42]	NA	T	Vibrations similar to heart rate
[65]	NA	T	Head vibrations
[69]	NA	V A	Virtual reality and an essence through a necklace
[43]	NA	A	Music
[45]	NA	V A	deep muscle relaxation and Imaginary visual activities using virtual reality
[72]	V A	T	Arm vibrations
[77]	V	V	Guided breaths
[46]	NA	A	Guided breaths
[51]	V	NA	Recognition of stress related to their activities
[70]	V A	V A	Breathing
[47]	NA	V A	Meditation and mindfulness
[64]	V	V	Breathe, eat, jump rope, close your eyes, dance, play music, paint, take photos
[52]	V	V	Diaphragmatic breaths
[67]	NA	O	Breathing
[53]	V	V	Shows graphs with stress and arousal levels, it is configurable by the user
[25]	V	V	Breathing
[71]	NA	A T V	Music and respiration
[62]	V	V	Relaxation
[56]	V	V	Virtual Reality
[57]	V	V	A long list of interventions are presented in this study
[66]	NA	V	Walking
[78]	NA	V O	Horticultural Therapy
[75]	V	NA	Vibrations similar to heart rate
[55]	NA	V	Nature Break
[47]	NA	V A	Guides to breathe, listen to music and positive messages and memories good times
[42]	NA	T	Vibrations
[58]	V	V	Persuasive message
[52]	NA	V A O	Virtual Reality and scent
[74]	NA	T	vibration
[76]	V	NA	Visualize the stress levels of the subject and their group

Note: V—visual, A—auditory and T—tactile, O—Olfactory, NA—Not Available.

**Table 2 healthcare-11-02369-t002:** Validated scales, questionnaires, or indexes used in the studies to measure stress, anxiety, emotions, depression, and cognitive load.

Questionnaire and Scales	Paper ID.
Anxiety subscale of the Teacher Behavior Assessment System (BASC)	[70]
Body Uneasiness Test (BUT)	[68]
Children Depression Inventory (CDI)	[68]
Covi Anxiety	[65]
Depression, Anxiety, Stress Scale (DASS)	[67]
Generated Anxiety Disorder (GAD)	[61]
Patient Health Questionnaire (PHQ)	[52,61]
Goal Attainment Scale (GAS)	[54]
Post Traumatic Check List 5 (PCL-5)	[54]
Beck Anxiety Inventory (BAI)	[54]
Beck Depression Inventory (BDI)	[47,54,62]
Flourish Scale (FS)	[54]
Nasa Task Load Index (NASA—TLX)	[26,44,48]
Geneva Emotional Wheel (GEW)	[48]
Short Stress State Questionnaire (SSSQ)	[48]
Big five—french version (BFI)	[48]
Perceived Stress Inventory (PSI)	[60]
Perceived Stress Scale (PSS)	[45,46,47,50,62,74]
Profile of Mood States (POMS)	[47,49]
Relaxation Rating Scale (RRS)	[59,78]
State-Trait Anxiety Inventory (STAI)	[42,47,52,55,62,74,75,78]
Brief Symptom Inventory (BSI)	[47]
Brief Fear of Negative Evaluation questionnaire (bFNE)	[42]
Depression, Anxiety, Stress Scale (DASS-21)	[52]
stress subscale	[52]
Warwick-Edinburgh Mental Wellbeing Scales (WEMWBS)	[52]
Difficulties in Emotion Regulation Scale (DERS)	[74]
Stress Response Index (SRI)	[76]
Acculturative Stress Scale (ASS)	[66]

**Table 3 healthcare-11-02369-t003:** Self-regulation during stress episodes.

Paper ID.	Wearable Type	Type of Experiment	Stressor	Intervention Time Length	Participants	Study Location	Number	Age Category	Outcome
[61]	torso-wear	not controlled	NA	56 days	adults	any place	14	adults	HRV $t_{\left(13\right)}=11.00,p<0.001$ + ✽
[26]	wrist-wear	not controlled	NA	2:30 h	students	any place	15	adults	Decrease in systolic BP −5.81% ns and diastolic BP by −1.93% ns *p* < 0.05 + ✽
[41]	wrist-wear	not controlled	NA	5 min	employees	NA	30	adolescents and adults	+ quantitatively assessed the user’s stress level.
[62]	torso-wear and head-wear	not controlled	NA	4 weeks	students	University Laboratory	89	adults	Pre-post reduction in stress p=0.019 + ✽ ●
[42]	wrist-wear and torso-wear	controlled	speech preparation	NA	NA	any adult	25	adults	Lower levels of anxiety $t_{\left(1,50\right)}=2.79,p=0.0007$ + ✽ ●
[63]	torso-wear	controlled	NA	NA	NA	NA	NA	NA	+ Reduced stress
[43]	wrist-wear	controlled	NA	NA	older adults	NA	NA	older adults	+ Level of anxiety and depression
[44]	wrist-wear, shoulder-wear, hip-wear and back-wear	controlled	8 tasks challenging motor and cognitive and motor skills	90 min	students	Laboratory	15	adults	Subjective relaxation $F_{\left(3,32\right)}=7.58,p=0.0004$ + ✽ ●
[45]	wrist-wear	controlled	exams and academic deadlines	3 weeks	students	NA	26	adults	+ reduce stress
[73]	arm-wear	not controlled	NA	6 h	any adult	work and house	2	NA	Effect of the alleviation activities F=7.72,p=0.003 + ✽
[46]	wrist-wear	controlled	NA	10 to 20 min every day for four weeks	any adult	laboratory	35	adults	Reduction of stress $t_{\left(28\right)}=-4.925,p<0.001$ + ✽ ●
[74]	arm-wear and eye-wear	not controlled	NA	2 weeks	students	laboratory	39	adults	Perceived Stress $F_{\left(1,37\right)}=25.65,p=0.000$ - ✽ ●
[64]	torso-wear	controlled	classroom activities	36 h	students	classroom	2	children	+ stress balance
[47]	wrist-wear	controlled	Stroop-like task	4 weeks	any adult	laboratory	55	adults	Perceived stress $t_{\left(45\right)}=-3.609,p=0.001$ + ✽ ●
[76]	hand-wear	controlled	NA	3 weeks	unemployed	laboratory	62	adults	Regulating function and coping ability *p* < 0.017 + ✽
[65]	torso-wear	controlled	video games	10 min	any adult	NA	10	adults	A + increase self-awareness, support social interactions, and give back
[69]	head-wear	controlled	NA	5 min	any adult	neutral office space	12	adults	Relax scores increased significantly $t_{\left(11\right)}=-3.609,p=1.43$ + ✽
[71]	head-wear	not controlled	NA	5 days	any adult	any place	7	adults	+ † A effective in helping users cope with anxious states
[72]	neck-wear	controlled	Sing-a-Song Stress Test	15 seg × task	students	laboratory	8	adults	Change in HR + ✽ ●
[48]	wrist-wear	controlled	difficult mode of game, time pressure action	50 min	Social network users	any place	29	adults	- ✽ ● physiological pBonferroni<0.01 subjective, there was no effect of the biofeedback behavioral no differences were found
[50]	wrist-wear	controlled	lost of mobility due COVID-19	1 h	students	university	24	adults	- ✽ ● stress relieving F=28,p<0.01
[75]	torso-wear and fingers-wear	controlled	compound remote associate (CRA) task	NA	students	university	44	adults	efficacy to lowered their breath rate p<0.0001
[49]	finger-wear	not controlled	NA	120 min	any adult	forest	48	adults	✽ ● decrease negative mood states, HR and temperature p<0.001
[60]	torso-wear and wrist-wear	controlled	writing emails	7 min	workers	office	53	adults	✽ ● The VAS value, decreased from 4.81 to 1.02 (*p* < 0.001), and the PSI score also decreased from 16.75 to 10.60 p<0.001

Note: + Positive results, - Unsuccessful results ✽ Statistical analysis, ● Control Group, †—user experience, NA—Not Available.

**Table 4 healthcare-11-02369-t004:** Self-regulation therapies.

Paper ID.	Wearable Type	Type of Experiment	Stressor	Intervention Time Length	Participants	Study Location	Number	Age Category	Outcome
[70]	head-wear	controlled	NA	2 weeks	students	school	20	children	Within group, the intervention group improve significantly Calm score p=0.010 + ✽ ●
[67]	torso-wear	controlled	Stroop Test	3 min	graduate and undergraduate students, researchers, and employees	university	7	adults	Perceived long-term stress p=0.005 + ✽
[54]	wrist-wear	controlled and not controlled	NA	2 to 4 weeks	veteran	clinical and natural environments	14	adults	+ ✔ ❏
[55]	wrist-wear	not controlled	NA	10 min	students	NA	14	adults	No significant decrease in participants’ HRs p=0.0852 - ✽ ●
[56]	wrist-wear and eye-wear	controlled	academic tasks	NA	students	laboratory	2	adults	+ reduces stress level
[68]	torso-wear and finger-wear	controlled	Olfactory identification test	12 weeks-2 sessions week	Adolescents with Anorexia Nervosa	Clinic	6	adolescent	+ Aceptability, feasibility and use patterns selfreports of welbeing p=0.005

Note: + Positive results, - Unsuccessful results, ✽ Statistical analysis, ● Control Group, NA—Not Available, ✔—Helpful, ❏—Easy of use.

**Table 5 healthcare-11-02369-t005:** Awareness for prevention.

Paper ID.	Wearable Type	Type of Experiment	Stressor	Intervention Time Length	Participants	Study Location	Number	Age Category	Outcome
[51]	wrist-wear	not controlled	NA	15 days	social network users	any place	43	adults	Stress awareness p=0.02 + ✽ ● > U T
[52]	wrist-wear shoulder-wear, hip-wear and back-wear	not controlled	NA	4 weeks	students	university	132	adults	Reduce anxiety p=0.001 + ✽ ●
[53]	wrist-wear	not controlled	NA	95 min	professors	school	21	adults	+ A increased their self-awareness of arousal-related patterns
[77]	waist-wear	not controlled	NA	3 days	autistic students	school-based transition program	5	adults	+ calm and focused respiration patterns
[25]	torso-wear	not controlled	NA	35.7 min	office workers	any place	169	adults	Negative instance of stress p=0.002 + ✽ ●
[66]	torso-wear	not controlled	NA	24 weeks	migrant women workers	any place	132	adults	+ ✽ acculturative stress significantly decreased p=0.018 adherence, depression and acculturative stress
[58]	wrist-wear	controlled	Classroom activities	60 min	students	university	7	adults	+ positive results Awareness
[59]	wrist-wear and torso-wear	controlled	NA	1 min	NA	NA	16	adults	✽ Relaxation p<0.01
[78]	torso-wear	not controlled	Office work	4 h × 5 days	office workers	Office	24	adults	visualization information was easy to perceive, clear to understand, and was not interrupting, in general

Note: + Positive results, ✽ Statistical analysis, ● Control Group, >—User motivation, U—User engagement, T—Task Efficiency, NA—Not Available.

## Data Availability

Not applicable.

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
