# Peer review of "Wearables for Stress Management: A Scoping Review"

_healthcare, 2023, doi:10.3390/healthcare11172369_

Round 1

Reviewer 1 Report (Previous Reviewer 1)

For a review paper, there needs be an extensive literature review. This paper lacks that. I know so many researchers who have had significant contributions for stress detection and wearables for stress but those seemed to be missed by the authors. Sample researchers - https://smohanty.org/Publications.html, https://sites.google.com/view/laavanyarachakonda/publications-presentations, https://scholar.google.com/citations?user=brMG2D0AAAAJ&hl=en

Author Response

We thank the Editor for the opportunity to make major changes to our article, and the reviewers for taking the time to give us valuable feedback. We then address each point of the comments.

Reviewer 1

For a review paper, there needs to be an extensive literature review. This paper lacks that. I know so many researchers who have had significant contributions for stress detection and wearables for stress but those seemed to be missed by the authors. Sample researchers - https://smohanty.org/Publications.html, https://sites.google.com/view/laavanyarachakonda/publications-presentations, https://scholar.google.com/citations?user=brMG2D0AAAAJ&hl=en

Considering the methodological approach of a scoping review, we established inclusion criteria that allowed us to identify studies to answer our research question. Therefore, studies that did not meet these criteria were excluded. For instance, we did not consider studies that merely present a method to detect stress levels but instead focused on those presenting an evaluation of technology to understand its impact related to health outcomes or technology’s usability.

Finally, it is essential to point out that our methodological approach could have influenced the extensibility of the article to which the reviewer refers. Therefore, we briefly describe the differences between the existing methodological approaches for a literature review. There are different types of literature reviews, for example, survey (Brena et al., 2017), rapid literature review (Klerings et al, 2023), mini review (Munawar et al., 2021), systematic review (Pollock and Berge, 2018), scoping review (Tricco et al, 2015), to mention a few. Their differences lie in their objectives, methodological approaches, and scope. This allows researchers to address different research questions at varying levels of depth. In our contribution, we propose a scoping review, which focuses on mapping the existing literature on a specific topic and identifying the extent and scope of available evidence. In contrast, a survey summarizes the related literature, which may be identified without necessarily following a structured methodological approach. At the same time, a systematic review seeks to synthesize and evaluate the quality of experimental studies to include those presenting a high level of evidence.

References

Brena, R. F., García-Vázquez, J. P., Galván-Tejada, C. E., Muñoz-Rodriguez, D., Vargas-Rosales, C., & Fangmeyer, J. (2017). Evolution of indoor positioning technologies: A survey. Journal of Sensors, 2017.

Klerings, I., Robalino, S., Booth, A., Escobar-Liquitay, C. M., Sommer, I., Gartlehner, G., ... & Waffenschmidt, S. (2023). Rapid reviews methods series: Guidance on literature search. BMJ Evidence-Based Medicine.

Tricco, A. C., Antony, J., Zarin, W., Strifler, L., Ghassemi, M., Ivory, J., ... & Straus, S. E. (2015). A scoping review of rapid review methods. BMC medicine, 13(1), 1-15.

Pollock, A., & Berge, E. (2018). How to do a systematic review. International Journal of Stroke, 13(2), 138-156.

Munawar, N. A., Hadiaty, F., & Parantoro, A. (2021). Business Network Literature Review: A Mini-Review Approach. Dinasti International Journal of Economics, Finance & Accounting, 2(1), 46-54.

Reviewer 2 Report (New Reviewer)

The purpose of the reviewed manuscript was to analyze wearable devices for stress management: a scope review

The topic is extremely interesting and it is believed that some points of interest can be suggested to the authors.

- The reference list is not accurate and in many cases there is no DOI, ad in 16,17,19 etc... 10.1109/ACCESS.2021.3085502

- The authors wrote that there are no other systematic reviews on the same subject but there are no references to the PROSPERO subscription, can they explain why?

- The PRISMA diagram is not shown exactly in fig.1

There are no other major doubts, an explanation on the points indicated is awaited

Author Response

We thank the Editor for the opportunity to make major changes to our article, and the reviewers for taking the time to give us valuable feedback. We then address each point of the comments.

Reviewer 2

The purpose of the reviewed manuscript was to analyze wearable devices for stress management: a scope review

The topic is extremely interesting and it is believed that some points of interest can be suggested to the authors.

- The reference list is not accurate and in many cases there is no DOI, ad in 16,17,19 etc... 10.1109/ACCESS.2021.3085502

DOI has been added to the references.

- The authors wrote that there are no other systematic reviews on the same subject but there are no references to the PROSPERO subscription, can they explain why?

PROSPERO is an openly accessible online repository comprising systematic review protocols covering a diverse array of subjects. As this database primarily displays protocols for reviews that are still in progress, we decided e to search for literature in databases commonly employed for conducting literature reviews, such as Scopus. The purpose was to juxtapose them with our own work and demonstrate our unique contribution. We have mentioned the databases used for the replicability of this study. However, we understand the reviewer's concern, so in the Discussion section (4.3, third paragraph) we already recognized that a limitation of our study is that the search was not exhaustive, because other databases, such as PROSPERO, were not included.

A diagram is not shown exactly in fig.1

The flowchart of the review is based on the PRISMA 2020 template for flow diagrams in new systematic reviews, which includes searches of databases and registers only, available at: http://www.prisma-statement.org/PRISMAStatement/FlowDiagram?AspxAutoDetectCookieSupport=1. This template includes the stages of identification, screening, and inclusion; which are presented in our diagram. 

There are no other major doubts, an explanation on the points indicated is awaited

Reviewer 3 Report (New Reviewer)

The paper is ready for publication, it's quite complete and comprehensive. I would slightly improve the language. 

The language just need minor improvements in terms of fluency (especially for sentences length).

Author Response

Reviewer 3

The paper is ready for publication, it's quite complete and comprehensive. I would slightly improve the language. 

The language just need minor improvements in terms of fluency (especially for sentences length).

The manuscript has been thoroughly reviewed to identify and correct grammatical and spelling errors, aiming to enhance the quality of the text and ensure it is written accurately in English.

Reviewer 4 Report (New Reviewer)

I read the article, which fits within the scope of this Journal, with considerable interest. In my personal view, the topic is intriguing enough to pique the readers' interest. The text is well-structured, and while much of it is clear, there are no odd sentence forms. The overall level of quality improves the survey's readability. While English is generally utilized appropriately in terms of grammar and syntax. The main contribution addresses gaps that might otherwise lead to new, demanding, and substantial research avenues.

The abstract concisely outlines the subject, method, and findings from the review

The authors in the introduction declared their goal, introduced their vision, and concretized their way of thinking. They also focused on the meaning of wearable devices (or wearables) and the challenges of their work and the same time made clear what a scoping review is and concluded to the research questions.

In Section 2 they presented in detail how a scoping review was carried out following the requirements of the Preferred Reporting Items for Systematic Reviews and Meta-Analyses extension for Scoping Reviews (PRISMA).

They developed their keyword strategy, listed the databases they searched in that particular time period, and were very detailed about their inclusion and exclusion criteria. They then recorded the collection study with Rayyan software and why. In the final subsection of section 2 the authors examined data extraction and analysis followed deductive and inductive approaches.

The results were presented clearly and analyzed appropriately

In the “Discussion and Limitations” section the authors discussed the results of their research by relating them to the existing literature. The section also highlighted the possibilities of practical application in the future, with research and in which directions while setting the limitations of this work.

The Conclusions section was supported by the results and also strengthened the plan for further work.

Author Response

We thank the Editor for the opportunity to make major changes to our article, and the reviewers for taking the time to give us valuable feedback. We then address each point of the comments.

Reviewer 4

I read the article, which fits within the scope of this Journal, with considerable interest. In my personal view, the topic is intriguing enough to pique the readers' interest. The text is well-structured, and while much of it is clear, there are no odd sentence forms. The overall level of quality improves the survey's readability. While English is generally utilized appropriately in terms of grammar and syntax. The main contribution addresses gaps that might otherwise lead to new, demanding, and substantial research avenues.

The abstract concisely outlines the subject, method, and findings from the review

The authors in the introduction declared their goal, introduced their vision, and concretized their way of thinking. They also focused on the meaning of wearable devices (or wearables) and the challenges of their work and the same time made clear what a scoping review is and concluded to the research questions.

In Section 2 they presented in detail how a scoping review was carried out following the requirements of the Preferred Reporting Items for Systematic Reviews and Meta-Analyses extension for Scoping Reviews (PRISMA).

They developed their keyword strategy, listed the databases they searched in that particular time period, and were very detailed about their inclusion and exclusion criteria. They then recorded the collection study with Rayyan software and why. In the final subsection of section 2 the authors examined data extraction and analysis followed deductive and inductive approaches.

The results were presented clearly and analyzed appropriately

In the “Discussion and Limitations” section the authors discussed the results of their research by relating them to the existing literature. The section also highlighted the possibilities of practical application in the future, with research and in which directions while setting the limitations of this work.

The Conclusions section was supported by the results and also strengthened the plan for further work.

Dear reviewer, we are grateful for your support and for recognizing the value of our work.

Reviewer 5 Report (New Reviewer)

The scoping review you are presenting about stress management with help of wearable devices is interesting, well conducted and data is well analyzed. 

Nevertheless, the presentation of the results and, above all, the discussion of the article has to be further improved.

Furthermore, there are some structural issues, tables are not well described and not in the right order, so it is difficult to follow your presentation and extract the findings. 

In detail, you should revise the following parts:

l. 202: "A total of 40 articles met the inclusion criteria from which we extracted data (see Figure 2)." - state what is shown in Fig. 2 or better do not cite the figure here as it belongs to 3.2

l. 204: explain here that fig. 2 contains all 40 results orderd by, showing the evolution of...

3.3.1. 

Table 1 and 2 are not mentioned in the text. They should be described before 3, 4 and 5

l. 213: clarify what you mean by "the form factor of the wearable"

l. 214, 222, 291: Tables are mentioned in a wrong order

I propose to separate the description of each table: first explain that you created those 3 tables and why, afterwards add the numbers. But do not derive conclusions here, this should only be made in the discussion part (e.g. "we show that...")

Furthermore, tables 3, 4 and 5 belong to sections 3.5.1., 3.5.2. and 3.5.3 and should be presented and described there. 

You should probably consider a reordering of the sections to describe the results in a logical order.

l. 214/215: unclear sentence:  "we show that the user’s body part, the most commonly used wearable devices are those worn on the wrists"

Figure 3 would be clearer if you added a pie chart in the centre, it is not clear what the body separated into two half parts means

3.3.3

l. 265: one column of Table 1 is cited, but the tables have still not been presented or described, it is though complicated to follow your analysis

The title of table 1 must be wrong, please revise it. Cite and describe the table in the text

4.1 the discussion is far too short and lacks deepness. You should discuss some more aspects of the data found, and contrast the different types of wearables, their usefulness in different situations etc. Discuss all sub-points analyzed in the results and focus on the three types of intervention you mention in the beginning. Refer to your research questions.

Conclusions

l 450/451: I think you mean that the results are NOT generalizable, rephrase the sentence to make it clearer, please

I cannot find a clear answer to your research question "How do wearables help people manage stress?"

Author Response

We thank the Editor for the opportunity to make major changes to our article, and the reviewers for taking the time to give us valuable feedback. We then address each point of the comments.

Reviewer 5

The scoping review you are presenting about stress management with help of wearable devices is interesting, well conducted and data is well analyzed. 

Nevertheless, the presentation of the results and, above all, the discussion of the article has to be further improved.

Furthermore, there are some structural issues, tables are not well described and not in the right order, so it is difficult to follow your presentation and extract the findings. 

In detail, you should revise the following parts:

  1. 202: "A total of 40 articles met the inclusion criteria from which we extracted data (see Figure 2)." - state what is shown in Fig. 2 or better do not cite the figure here as it belongs to 3.2

In line 202, the cite to figure 2 was removed. 

  1. 204: explain here that fig. 2 contains all 40 results ordered by, showing the evolution of...

A paragraph  (lines 203-205) has been added to describe the content of Figure 2.

3.3.1. 

Table 1 and 2 are not mentioned in the text. They should be described before 3, 4 and 5

Tables were mentioned in text: table 1 in line 278 (Table 1) and table 2 in  line 310. Further, the tables are described in order of appearance.

  1. 213: clarify what you mean by "the form factor of the wearable"

We modified the text to eliminate the phrase "the form factor of the wearable" and clarify the objective of the image.

  1. 214, 222, 291: Tables are mentioned in a wrong order

I propose to separate the description of each table: first explain that you created those 3 tables and why, afterwards add the numbers. But do not derive conclusions here, this should only be made in the discussion part (e.g. "we show that...")

Furthermore, tables 3, 4 and 5 belong to sections 3.5.1., 3.5.2. and 3.5.3 and should be presented and described there. 

You should probably consider a reordering of the sections to describe the results in a logical order.

We added a description of the tables and also rearranged them:s Table 1 description starts at line 266, table 2 description is at line 317, and tables 3, 4 and 5 description is at lines 334 to 338.

  1. 214/215: unclear sentence:  "we show that the user’s body part, the most commonly used wearable devices are those worn on the wrists"

It was changed as follows: “ In Figure 3 it can be seen that wrist-worn devices (N=35, 87.50%) are the most commonly used among users…”

Figure 3 would be clearer if you added a pie chart in the centre, it is not clear what the body separated into two half parts means

The figure 3 has been modified to remove the divided body, as it did not have a meaningful representation. Additionally, the wording has been revised to provide an explanation of the image (lines 214 to 218).

3.3.3

  1. 265: one column of Table 1 is cited, but the tables have still not been presented or described, it is though complicated to follow your analysis

The title of table 1 must be wrong, please revise it. Cite and describe the table in the text

Thank you for your observation. The table title has been modified,  and  we cite and describe it on the text from  lines 263 to 268 .

4.1 the discussion is far too short and lacks deepness. You should discuss some more aspects of the data found, and contrast the different types of wearables, their usefulness in different situations etc. Discuss all sub-points analyzed in the results and focus on the three types of intervention you mention in the beginning. Refer to your research questions.

We made changes in Discussion (lines 420 to 451) to address your comments.

Conclusions

l 450/451: I think you mean that the results are NOT generalizable, rephrase the sentence to make it clearer, please

The conclusions section was modified in lines 489 to 497 to provide a clearer explanation. 

I cannot find a clear answer to your research question "How do wearables help people manage stress?"

We made changes in sections 3.2 (lines 205 to 210), 3.3.1 (lines 213 to 216), 3.3.3 (lines 266 to 272), 3.4.1 (lines 317 to 323), 3.5 (lines 334 to 337), 4.1 (lines 420 to 451), and 5 (lines 487 to 520) to revisit the research question and respond to it. We emphasized that our results provide evidence that the studies demonstrate how wearables can be used to help people through three strategies for stress management: 1) self-regulation during stress episodes, 2) self-regulation therapies, and 3) raising awareness for stress prevention.

Round 2

Reviewer 2 Report (New Reviewer)

The answers of the authors have been exhaustive and I have no negative evaluations

Reviewer 5 Report (New Reviewer)

Thank you for addressing the issues I detected, I have no further comments on that manuscript. 

This manuscript is a resubmission of an earlier submission. The following is a list of the peer review reports and author responses from that submission.

Round 1

Reviewer 1 Report

For a review, this is significantly less work. I like the writing style and the diagrams but a more vigorous search could have been done, for example, I know some scholars in the area who have contributed significantly to the area who were not cited, and their work was not mentioned. For example, here is one of the many papers that should have been mentioned and analyzed: https://ieeexplore.ieee.org/document/8830476 

Reviewer 2 Report

Dear Authors,

I have reviewed your manuscript titled "Wearables for Stress Management: A Scoping Review" submitted to the Journal of Healthcare. Your manuscript addresses a well-defined and interesting question, focusing on the effectiveness of wearable devices in stress management through a scoping review. 

The manuscript is clear, relevant to the field, and well-structured. The cited references are mostly recent publications and relevant to the topic. The figures, tables, images are appropriate and properly show the data (except for the issues mentioned below), and they are easy to interpret and understand. The data is interpreted appropriately and consistently throughout the manuscript. The review is clear, comprehensive, and of relevance to the field.

The paper fits the scope of the Journal of Healthcare, as it pertains to the application of technology in managing stress, a crucial aspect of overall health. The topic of stress management using wearable technology is likely to attract a wide readership due to its current relevance in the intersection of digital health and mental well-being.

I recommended the acceptance of this paper in the Journal of Healthcare, subject to minor revisions. The following improvements should be addressed in the revised version:

1. Correct the typographical errors.

2. Improve the readability of figures, specifically Figure 3 and Figure 4, for better comprehension in black and white print versions. Consider using patterns within the bars in addition to or instead of colors to ensure clarity in distinguishing different types of wearables.

3. In Reference 18, include "World Health" before "Organization": World Health Organization. COVID-19 Disrupting Mental Health Services in Most Countries, WHO Survey. Technical report, Genf, Schweiz, 2020.

4. For References 22 and 31, provide more information, such as the report title, publisher, URL, and access date.

5. In Reference 23, include more information about the source (title, publisher, URL, access date) to complete the reference.

6. Expand the Limitations subsection. It is important to acknowledge the potential limitations beyond the language constraint in conducting a scoping review.

Sincerely,
Reviewer

The manuscript is written in an appropriate and understandable English language, although there are a few typographical errors that need to be corrected. For example:

- hearables ---> Wearables (Page 2, Line 82)
- such as health condition --> such as health conditions (Page 8, Line 298)
- movements in legs --> movements in the legs (Page 14, Line 379)

Reviewer 3 Report

This is an exciting review of wearable devices for stress management. With technological advancement, it is critical to collate evidence on the role and effectiveness of these devices in assisting stress self-management. The authors did a great job in synthesising data in this scoping review considering various factors such as tools used and various outcomes measured. I enjoyed reviewing this work. The manuscript yet can be improved to increase rigor and transferability, facilitate an understanding of how included studies were conducted and the interpretation of their results.

One main comment is that in this manuscript, either process details are not clear or authors provide process details after discussing findings; this is very confusing. For example, the authors introduce tables 2-4 (line 173) and do not specify how studies are classified into 3; later, on page 8, the authors stated 3 classes based on the study objective, which implies that each table was relevant to one of these objectives. This should have been specified.

The authors specified that they follow guidelines for Scoping Reviews. One of the critical items for reporting reviews is following PICO (participants, Intervention, Comparator, Outcome). For example, the author missed the comparator element throughout. PICO should be followed for all steps, e.g., specifying eligibility criteria, and results. I suggest the Authors use a checklist to ensure they report all required details.

I wonder what authors’ justification is for conducting a scoping review, not a systematic review. Scoping reviews are for mapping literature and identifying gaps so they are generally done in a broad area of interest (e.g. stress management with digital tools). It also includes a wide range of databases. While systematic reviews look at the impact of a specific intervention (like this review of wearables). To me, it seems that the topic is a good candidate for systematic reviews to assess the impact, as interventions are all about wearable devices here.

Detailed comments:

Abstract: line 7 ‘some studies’ please provide value, how many? Line 8 again, how many studies and what was the study design?

Introduction:

Line 52 ‘hearbales’? Or wearables? I also suggest investigating if using the word “wearables” is a correct scientific method of refereeing to ‘wearable tools’. It seems more like an informal word.

Line 79, please provide a reference for “previous reviews”

Line 92-94 What is the justification of authors to classify interventions based on 3 types? Is this done based on existing literature, theory, a framework for stress management interventions or a classification theme based on the identified studies?

Methods:

Line 100: IEEE Xplore is already covered by the Scopus database, so there actually are only two databases. Here is a quote from the IEEE Xplore website. "The IEEE provides SCOPUS with all of the IEEE Xplore digital library content so that the bibliographic information (what is seen on an abstract page in IEEE Xplore, including the abstract) can be made visible.

Line 139: Data extraction process needs to be clear how many people conducted data extraction and whether there were any duplicates to ensure the validity and accuracy of the data extracted.

 Line 157: Why results were not extracted from the result sections of studies and only relied on the conclusion. Any misinterpretation from the study authors may be carried forward to this review. 

Did the authors exclude studies that measured interaction and adaptation (e.g. satisfaction) as an intervention outcome? It should be specified in the eligibility criteria as to what outcomes are considered and not considered.

 Results

Line 171: There is a need for a section to describe included studies’ design, e.g. whether the study had any controlled arm (number of study arms) and what the control arm included, e.g. nothing or passive intervention etc., if there were any before-after study or quasi-experimental? This information needs to be provided in the tables too.

Also, what is the length/duration of the interventions? Were they all lab best tests, then how long, how was the lab setting, and the background information is missing.

Line 179 Sentence ‘We found that 6 studies (20.68%) used a data acquisition system with more than one sensor worn in several body parts’ Please provide references for studies.

Line 183, ‘commercial software’, please provide details examples.

Line 190, Sentence ‘The studies used opportunistic and participatory sensing paradigms for collecting data from users’ The difference between these two is unclear; for example, heart rate is related to both methods. Please clarify. Were these paradigms impacted outcomes or device use/adherence differently in studies?

Figure 4: What is the difference between the classification of 'sensors' in the figure legend. I assume other wearable tools all contain sensors. Is it referring to studies that used 'multiple sensors'?Please clarify.

Line 240, the sentence is confusing: In the text, it says experiments run under controlled conditions' yet in the referred tables, there are studies classified as uncontrolled; please clarify. Also, what does ‘controlled (NA)’ mean in tables 2-4?

Line 256, ‘Stressor’ and ‘controlled condition’ need to be elaborated a bit more.

Line 258-263, Are these lines related to ‘outcome measures’? It does not provide further details about outcome measures other than stress. Or are these lines related to the tools used? Then what tools were used by the remaining studies? This isn't very clear. Maybe combine it with the next section of ‘study outcomes’, after restructuring it in a way that it is clear A) what are all outcome measures, B) what time points were outcomes measured?  C) how have they changed after the experiment? The authors need to talk about efficacy more clearer, considering interventions and designs.

Table 1: Wouldn't it be better to format table 1, one way around? Meaning rather than listing what tools each study used, list tools and relevant studies next to it. This will prevent the list goes long, it also can show what tools are utilised commonly and how the findings of studies are comparable considering the tools used.

The paragraph starting with line 290, Is this the intervention used in the study? The paragraph is written more like a discussion than an actual experiment; if so, then these points should be discussed in the discussion section.

Some tables (Tables 2-5) appear in the middle of the discussion section. Tables need to appear either at the end of the manuscript or the closest place where they are referred to.

 Tables 2-4:  What does the type of experiment column try to convey by ‘controlled (NA)’, if controlled, the control condition should be elaborated consistently. Formatting issues are noted; the participants, and location columns are mixed up throughout the three tables.

 Discussion

Line 382-384: Please provide references for your statements and why these elements are important. The proposed idea hasn’t been fully elaborated to aid readers.

 Line 383, limitations: The authors used only used two databases why relevant databases such as Embase and Psychinfo are not included? No quality assessment was done for the included studies. These should be specified in the limitation section.

Reviewer 4 Report

The manuscript with the title “Wearables for Stress Management: A Scoping Review” (healthcare-2384650) is well written.

The introduction has a clear structure, derived research questions are appropriate. Methods including study design, selection and analysis as well as the presentation of results are sound and clearly presented. Discussion including limitations and conclusion are also clearly presented. The first part of the discussion seems too short and not sufficiently related to the results.  I would suggest to improve this.

In the following I have some comments / further suggestions:

(1) Line 92/93: “Self-regulation during a stress episode” and “Self-control during stress episodes.” seems pretty similar to me. What is the exact difference? You describe self-control based on long-term goals, but the selected studies under 3.5.1 and 3.5.3 are partly quite similar (biofeedback in the broadest sense). In my opinion “self-regulation” in a corresponding stress situation to regulate in this moment should be better differentiated from “self-control” in terms of long-term goals (a different term could be helpful as well). Overall, I do not have the impression that these constructs are well differentiated.

(2) Table 2: it would be helpful to provide information towards effect sizes (maybe calculated into Cohen´s d for better comparison). I checked paper ID 51 reported in table 2 for interest reasons.  In the process I have found on purpose that this study analyzed 98 subjects and not 89 which you have reported. I do not have the capacity to check all results, please control carefully for mistakes. ID45 in table 2 is not significant (p=1.43).

(3) line 370: pet society? I do not understand how this fits into the framework of the article.